# Insecticides, more than herbicides, land use, and climate, are associated with declines in butterfly species richness and abundance in the American Midwest

Braeden Van Deynze[1]*, Scott M. Swinton[2], David A. Hennessy[3], Nick M. Haddad[4], Leslie Ries[5]

1 Washington Department of Fish and Wildlife, Olympia, WA, United States of America, 2 Department of Agricultural, Food, and Resource Economics, Michigan State University, East Lansing, MI, United States of America, 3 Department of Economics, Iowa State University, Ames, IA, United States of America, 4 Kellogg Biological Station and Department of Integrative Biology, Michigan State University, Hickory Corners, East Lansing, MI, United States of America, 5 Department of Biology, Georgetown University, Washington, DC, United States of America

* braeden.vandeynze@dfw.wa.gov

**Data Availability Statement:** The data and metadata are on figshare: https://doi.org/10.6084/m9.figshare.25857424.

## Abstract

Mounting evidence shows overall insect abundances are in decline globally. Habitat loss, climate change, and pesticides have all been implicated, but their relative effects have never been evaluated in a comprehensive large-scale study. We harmonized 17 years of land use, climate, multiple classes of pesticides, and butterfly survey data across 81 counties in five states in the US Midwest. We find community-wide declines in total butterfly abundance and species richness to be most strongly associated with insecticides in general, and for butterfly species richness the use of neonicotinoid-treated seeds in particular. This included the abundance of the migratory monarch (*Danaus plexippus*), whose decline is the focus of intensive debate and public concern. Insect declines cannot be understood without comprehensive data on all putative drivers, and the 2015 cessation of neonicotinoid data releases in the US will impede future research.

## Introduction

Like many other insects, butterfly populations are showing widespread signals of decline [1]. Globally, declines in abundance and biomass for many insect groups have been reported at rates of 2–4% per year, rates that compound over two or more decades to as much as 30–50% loss of total abundance [2, 3]. For butterflies, declines have been widely reported, including in the UK, Netherlands, and grassland habitat across Europe [4]. In the United States, widespread declines have been reported, including in the Midwest [5], with similar results using independent data in Ohio [6], and Illinois [7], but with the starkest declines in the Western US [8, 9]. Exceptions to the general pattern of insect decline include no trends for aquatic insects globally, but this was controversial [10]. Moths In the UK have also shown no trends [11]. Given

**Funding:** NSF Long-term Ecological Research Program grant DEB 2224712 (SMS, BVD, NMH) Michigan State University AgBioResearch (SMS) USDA National Institute of Food and Agriculture (SMS) Elton R. Smith Endowment for Food and Agricultural Policy (DAH) USGS Midwest Climate Science Center fund: Grant No. G21AC10369- 00.

**Competing interests:** The authors have declared that no competing interests exist.

that the majority of studies on terrestrial insects indicate either declines or lack significant trends, this suggests that, taken in total, overall declines are overwhelmingly supported by the evidence. With such steep losses comes evidence for a decline in the many ecosystem services provided by insects [12–14] and the mounting risk of ecosystem collapse [15]. Present-day bird declines in Europe have been linked directly to pesticide use [16], and the recent, estimated loss of three billion birds in America north of Mexico [17] has also been hypothesized to be linked to insect losses [18, 19].

Despite recent widespread insect declines, evidence on the primary *drivers* of these declines remains murky [1]. The three most widely recognized global drivers of insect declines are land conversion, climate change, and agricultural pesticides (insecticides and herbicides). Yet the relative impact of each has been shifting over time and also interacting in ways that make it hard to tease apart their individual contributions [20]. Further, these primary drivers have changed in importance over time. While land use conversion is generally recognized as the historical driver of most species loss, landscapes in North America underwent the most dramatic conversions decades before the onset of recent insect declines [21]. Although climate change acceleration over the last 20 years is now a primary focus of biodiversity loss [22], taxon-wide studies for insects show species respond variably, depending on their thermal niche requirements, producing climate "winners" and "losers" [23].

In contrast, agricultural chemical applications are, alone, the only putative driver of herbivorous insect declines specifically formulated to be either directly lethal to insects (insecticides) or to reduce the cover of plant species (herbicides) in the immediate or surrounding landscape on which many insects depend [20]. Recent US and Canadian studies have shown a negative relationship between pesticides and abundance of monarchs (*Danaus plexippus*) [24–26], dragonflies (*Odonata*), and bumble bees (*Bombus occidentalis*) [27, 28], and lowland butterflies in agricultural regions of central California [29]. In large-scale studies in the UK, agricultural applications of seed-coated neonicotinoids were found to be associated with decreased abundances of bees [30] and butterflies [31].

Quantifying insecticide and herbicide applications (hereafter, "pesticide" when referred to jointly) and their effects over large scales and long time frames is particularly difficult (Box 1, Fig 1). This is because, unlike climate and land-use, where global time-series spatial data layers are publicly available at multiple scales, pesticide applications cannot be sensed remotely. US data on pesticide use, in coarsened form, have been publicly available for years, but difficult to access and use [32], but have recently been synthesized into large-scale ecological analyses of the impacts of pesticide use [33]. Unfortunately, data at this resolution fall short of demands of analyses that tease out impacts across all hypothesized stressors. A more granular version of data on pesticide use are available via proprietary data, such as detailed surveys of individual farmers and the types and mixes of chemical and seed technologies they use, which vary substantially, not only over time but on a farm-by-farm and regional basis [34–36].

## Box 1. An overview of pesticide use and modes of impact

The commercial release of genetically modified (GM) crops and insecticide seed coatings during 1996–2003 (Fig 1) sharply narrowed the set of pesticides used by most farmers, with repercussions on local biodiversity [37–39]. We provide an overview of different classes of pesticides, their potential impacts on butterflies, and their change in use over the period of this study.

Insects exposed to pesticide toxins can experience sub-lethal effects, lethal effects, or both. Insects can be harmed by weed control that decreases vegetative diversity [40] and reduces habitat, host plant, and floral resources [41–43]. We classified pesticides, including herbicides and insecticides, into six broad technological categories that represent major shifts since the 1990s and shape how different pesticides and pest control practices are likely to impact herbivorous insects 40]. Our 1998–2014 analysis period includes several important trends: rising adoption of glyphosate-tolerant and *Bt*-traited crop seed, introduction and rapid adoption of neonicotinoid-treated crop seed, and decline followed by increase in non-glyphosate herbicide use (Fig 2H–2J). The specific timing of adoption and disadoption of these practices varies over our study region, providing a "natural experiment" that allows for robustly teasing out the differential impacts on butterfly populations.

Insect control can be achieved via broadcast spray of chemical toxins or, more recently, via advances that incorporate toxins directly into plant tissue which target the insects most likely to feed upon them (Fig 1). Sprayed insecticides, represented by pyrethroids and organophosphates, are traditionally used reactively in response to insect infestations when they are detected. They pose a direct threat when non-target insects are exposed to spray or residues. We assign such insecticides to the "reactive" group. More recently, farmers have relied on prophylactic, systemic pest control, employed each year. These pest control methods rely on toxins incorporated directly into crop tissue through either chemical seed coatings containing neonicotinoids or insecticidal proteins introduced via genetic engineering, such as those produced by the Lepidopteran-specialist bacteria, *Bacillus thuringiensis* (Bt). Bt traited seed is genetically modified to produce insecticidal proteins. Such seed has been available for corn since 1996, with proteins targeting European corn borer, and since 2003, corn rootworm and earworm as well. We observe increasing adoption over the period of our study. With these methods, the incorporated toxins should only impact insects that feed directly on the targeted crops, including corn across the Midwest US. For Midwest US butterflies, no species uses corn as a host plant and only two species are known to feed on soybean (*Epargyreus clarus* and *Vanessa cardui*). Both are generalists using a wide variety of other host plants.

Separate from exposure via crop tissue or direct spray is exposure to insecticide residues that persist in the soil and water or that contaminate the tissue of non-target plants. Early concerns about contamination of pollen proved to be unfounded [44]. However, this issue has been especially concerning for neonicotinoids, which are often applied as a seed coating in the form of an insecticidal dust coating on corn or soybean seed that is taken up by plant tissue as the crop develops. Up to 85 percent of the coating can leach into the environment [45]. When neonicotinoids are present in plant tissue, they provide protection from a broad-spectrum of insect pests. Their use has grown dramatically since their introduction in 2004 coinciding with the appearance of soybean aphid and also to meet demand for additional systemic insecticides to supplement Bt-traited corn, whose target insects quickly evolved to resist [46]. Leached chemicals persist in the water and soil. Plants growing in that soil include forage or hosts for insects, including butterflies [37,47,48], and thus provide a substantial pathway for "targeted" chemicals to reach non-target species.

Weed control is most often achieved through herbicides, which are typically applied via broadcast spray during the crop season and are not known to cause acute harm to

insects. However, by killing weeds and other non-target plants near cropland, insect habitat and forage resources are reduced [49]. The major technological shift in weed control since the 1990s has been the rise and continued dominance of glyphosate herbicides, commonly marketed as "Roundup". Since the introduction of corn and soybean seed genetically engineered to tolerate this broad-spectrum herbicide, farmers have come to rely primarily on glyphosate for weed control in these crops. As a result, farmers increased glyphosate use while reducing the use of other herbicides (Fig 2J). This became particularly concerning for monarch butterflies since their host plants are strongly associated with row crops and their numbers began a sharp decline during the period of glyphosate adoption [50]. The near-exclusive use of glyphosate began to decline in 2008 (Fig 2J) as some weeds began developing genetic resistance [51]. During this period, farmers returned to applying multiple herbicides often targeted at specific weed species, resulting in shifting exposures of non-target insects to the effects of such chemicals on the environment.

The restricted spatial and temporal scope of many previous studies limited the ability to tease apart the effects of multiple pesticide classes from other environmental covariates, such as climate or land use, as we do in this study. There are two reasons why analyses that omit co-occurring drivers of insect decline or focus on a single pesticide active ingredient can be misleading or fail to detect associations. In the United States during 2004–2012, corn and soybean farmers rapidly adopted glyphosate, Bt genetics, and neonicotinoid-treated seeds while use of other pest control technologies declined due to substitution effects [36, 52]. As multiple pesticides may affect butterfly decline by one or more channels, each of these variables must be included if an analysis is to avoid omitted variable bias [53–55] or be useful for forecasting the potential impacts of pesticide regulations. Such comprehensive analysis requires substantial spatial and temporal coverage to tease apart substantially co-occurring drivers through natural variation over space and time (e.g., a "space-time substitution"). We use a panel dataset to overcome these limitations.

Here, we present a comprehensive analysis of the drivers of insect declines, incorporating the impacts of all three dominant drivers, land use, climate, and multiple pesticide use, in the Midwest (Fig 2). This region is the US center of corn and soybean production, with most counties in our study region having over 60 percent of land under agricultural production (Fig 2A–2C). The Midwest is also the US region with the densest network of butterfly monitoring surveys (Fig 2D). In our analysis of the major drivers of insect decline, we include all major classes of modern pesticides and group them into six categories based on their main active ingredient, degree of specificity, timing of applications, whether the treatment is applied prophylactically (every year) or reactively (in response to pest outbreaks), and the tendency of each to spillover and persist in the surrounding environment (Box 1, Fig 1).

We present a comprehensive analysis of the drivers of insect declines, incorporating the impacts of all three dominant drivers, land use, climate, and multiple pesticide use, in the Midwest (Fig 2). This region is the US center of corn and soybean production [56], with most counties in our study region having over 60 percent of land under agricultural production (Fig 2A–2C). The Midwest is also the US region with the densest network of butterfly monitoring surveys [57] (Fig 2D).

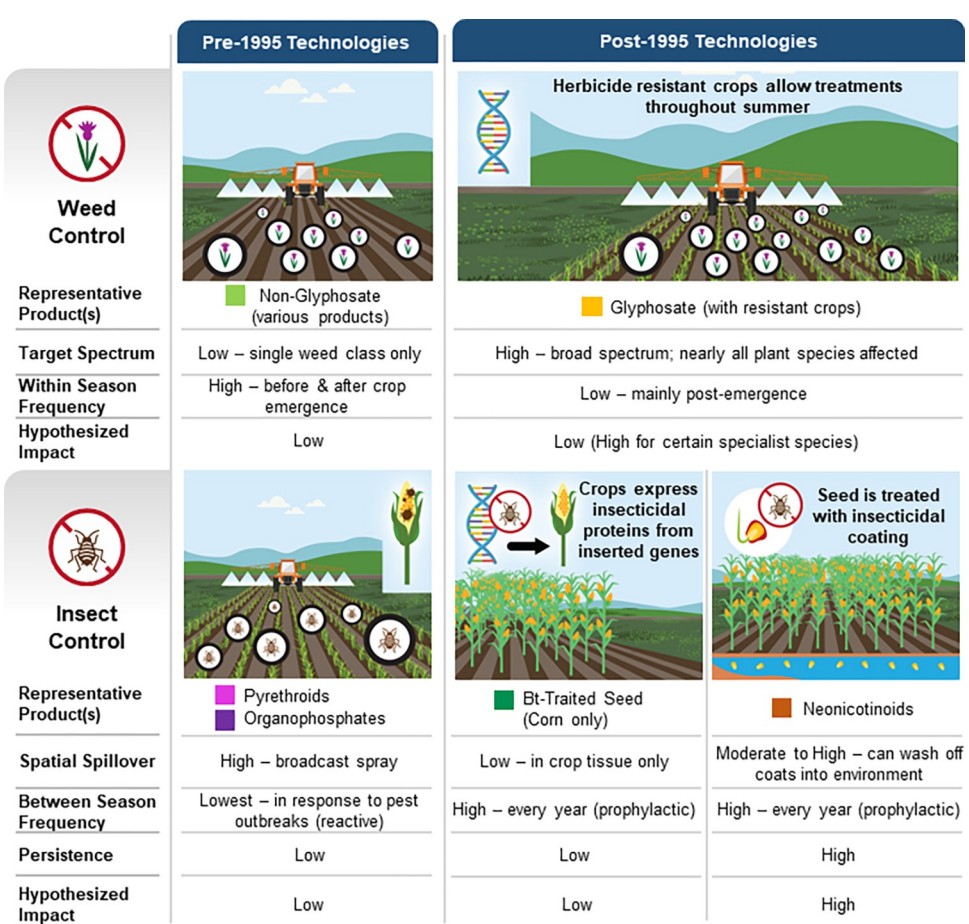

**Fig 1. Evolution of pest control technology in North American corn and soybean production.** Hypothesized impact refers to hypothesized impact on butterfly metrics.

## Methods

### Data construction

We bring together data from several sources to construct a unique panel dataset. The base geographic unit in the panel is a county and the base temporal unit is a year so the unit of observation is each county-year. The panel includes observations from 81 counties and 17 years (1998–2014). The annual number of monitored counties ranges from 15 to 37 based on data availability (Fig 2D). Counties entered the panel in different years as new monitoring programs were established (S1 Fig in S1 File).

### Butterfly abundance data

As a measure for butterfly abundance, the dependent variable in our analysis, we use county-year aggregates of monitoring surveys conducted by four volunteer programs associated with the North American Butterfly Monitoring Network [57]. Both the Illinois Butterfly Monitoring Network and the Ohio Lepidopterists provide data throughout the period of study, while the Iowa Butterfly Survey Network and Michigan Butterfly Network provide data from the starts of their survey programs in 2007 and 2011, respectively. Subsets of these data have been

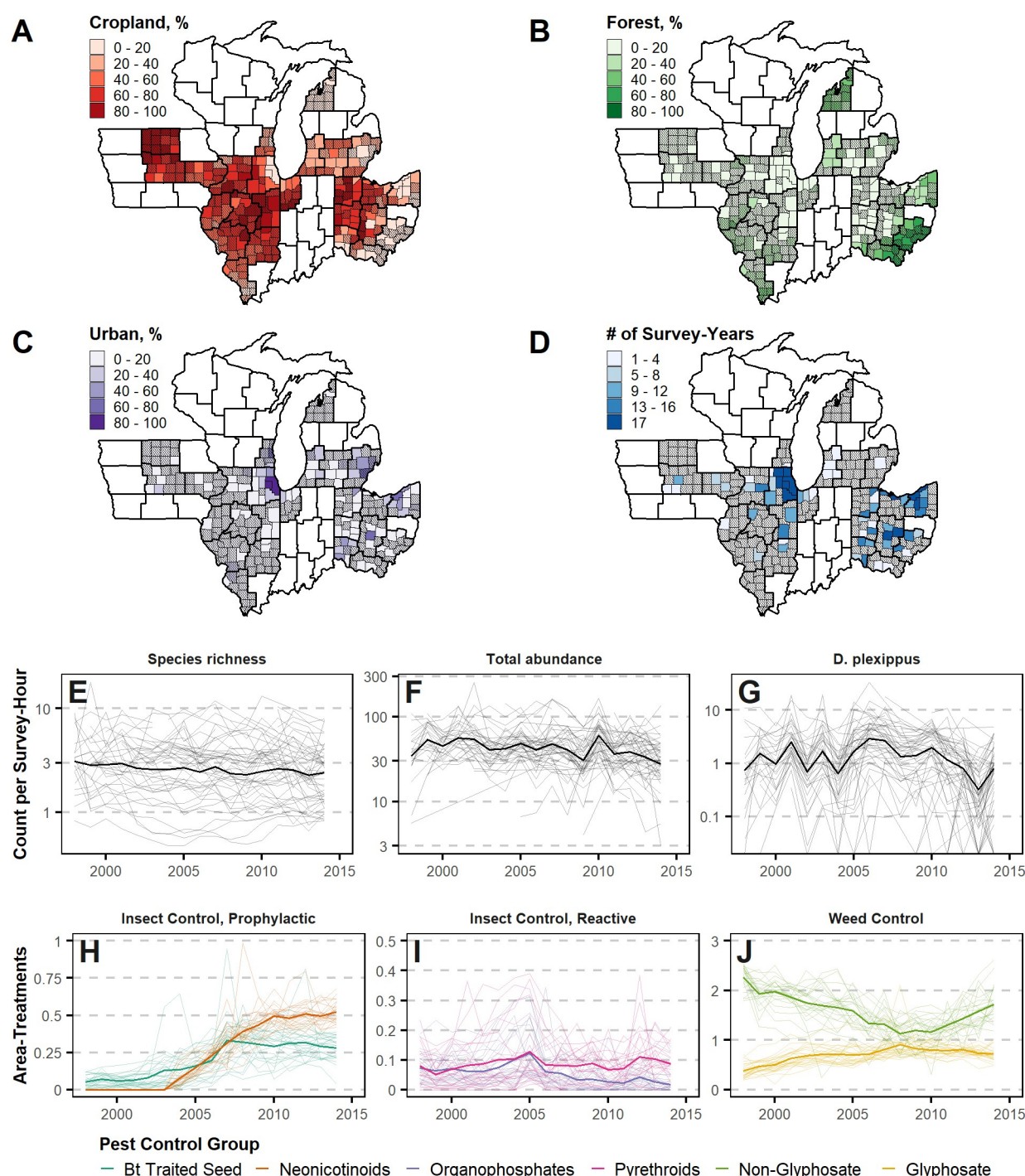

**Fig 2. Data summaries for key variables.** (A-C) shows land cover by county as percent of total land area in 2014 and (D) shows the number of years each county contributed butterfly population surveys; bold borders indicate USDA Crop Reporting District (CRD) boundaries while hatching shows counties not included in analysis due to lack of butterfly survey data availability. (E-G) shows the distributions of annual county-level species richness, total abundance, and monarch abundance measures; each line represents an individual county and the bold line represents the median value in each year. (H-J) shows area-treatments, the average number of times a pesticide or pest control treatment is applied to an acre of soybean or corn cropland, for each pest control technology group; each line represents a CRD and the bold line represents the median value in each year.

previously analyzed to assess overall butterfly population trends in Ohio [6, 58], and Monarch population trends in Illinois [25, 59].

Each butterfly monitoring network makes use of a common method known as Pollard surveys [60], which means that their protocols result in data sets that are appropriate to use in a unified analysis [57, 60, 61]. Routes are established by volunteers in coordination with network coordinators to (a) transect a variety of habitat types, (b) follow existing pathways so as not to disturb habitat, (c) be easily located by other volunteers, and (d) take between 30 minutes to two hours to complete. For each route, a single volunteer walks at a consistent pace along routes a minimum of six times per year during the months of June and July, with additional runs during these and earlier or later months when possible. Surveys are conducted between 10:00am and 3:30pm on days with (a) less than 50% cloud cover and (b) light to moderate winds. During the survey, the volunteer records all individuals by species sighted within roughly 5m to each side of the route. Volunteers are instructed to identify to species only with certainty and not to guess, but every individual observed is recorded [57].

Counts are summed across surveys conducted during June through August for each county-year. We analyzed the total number of species (Fig 2E) and abundance of all individuals across all species (Fig 2F). Aggregate counts of individuals identified by species across all county-years show a large number of species detected in the community with no one species dominating the total abundance measures (S2 Fig in S1 File).

Although our study's focus is on the response of the whole butterfly community, the effects that underlie the larger patterns are cumulative, integrated over all individual species. Accounting for individual responses requires grappling with the wide variety of life history details and ecological relationships that are unique to each species. While this analysis is beyond the scope of this study, we explored the response of a single species, the eastern migratory monarch butterfly (*Danaus plexippus*), in detail (Fig 2G). We chose this species because it has the highest public profile, is the most intensively studied butterfly species in the world [62], and is also under yearly review for listing under the Endangered Species Act [63]. Further, the Midwestern US produces proportionally more arrivals in the overwinter grounds than any other region of their breeding range [64], so our study region is particularly relevant to this species. The monarch's most common and preferred host plant, common milkweed, *Asclepias syriaca* [65], is closely associated with row crops and field margins, and thus is particularly vulnerable to agricultural practices [66].

### Land cover data

We measure land cover at the county-level as the proportion of land within each county under three land cover groups: forest, urban, and cropland (Fig 2A–2C). We use county-level pixel counts from the USGS National Land Cover Database, which classifies the entire land area of the conterminous United States into one of several land cover classes at 30m resolution. The database is published for six calendar years over the course of our panel: 2001, 2004, 2006, 2008, 2011, and 2013. For years when the NLCD was not published, we assign the value from the nearest available year using a rolling join, assigning the value from the nearest available year. The assignment of NLCD land cover classifications to land cover groups is described in S2 Table in S1 File.

### Pesticide use data

We used a unique, granular pesticide dataset, consisting of proprietary farm-level pesticide application data, including six primary pesticide classes defined for this study (Fig 2H–2J), collected via paid phone interviews by Kynetec USA, Inc., a market research company. These

data were collected via computer-assisted telephone interviews of soybean and corn growers. Lists of eligible growers were constructed from lists of growers who receive federal payments, membership lists of state and national growers associations, and subscription lists to agricultural periodicals. Sampling lists were constructed to ensure representativeness of applications at the level of the crop reporting district (CRD), USDA-designated groupings of counties in each state with similar geography, climate, and cropping practices. Non-respondents were recontacted at least eight times to reduce non-response error. Respondents were asked to detail their field-level pesticide, tillage, and seed choices during the previous growing season. Respondents were compensated monetarily upon survey completion while responses were crosschecked against realistic application rates and consistency with other reported practices. These surveys included neonicotinoid usage from 1998 until 2014 (Fig 2H), at which time surveys about this treatment were discontinued [67].

Pesticide use is measured in county-level area-treatments for each pesticide group. Area-treatments are defined as the average number of times a pesticide within a group is applied over a defined region (e.g., a field or a county) in a season [39]. Area-treatment measures are preferred over volumetric measures because their construction accounts for sometimes dramatic differences in application rates between different products [36, 39]. By accounting for both the geographic extent and frequency of application, area-treatments better capture the degree to which a butterfly population may have been exposed to a particular chemical. In precise terms, area-treatments are calculated for each pesticide group as the sum of soybean and corn acres treated on respondent farms within each CRD, divided by the total planted acres of soybean and corn in each CRD for each year. To capture county-level variation in exposure potential, we multiply the CRD-scale field-level area-treatments by the percent of land area identified as planted/cultivated land cover in the National Land Cover Database (NLCD). The resulting product is a county-level annual area-treatment estimate for each pesticide group of interest.

Farmers apply hundreds of distinct pesticide products to soybeans and corn. To simplify our analysis, we identify six groups of pesticides and crop protection technologies, divided into three classes, that together represent the majority and diversity of pesticide use on these crops (Fig 1). The first is weed control, represented by glyphosate and non-glyphosate herbicides. The second is insect control, reactive, which consists of the sprayed pesticide categories, pyrethroids, and organophosphates. The final class is insect control, prophylactic, which includes neonicotinoid seed treatments and Bt traited GM seed, two seed technologies that provide preventative protection from insects. The assignment of active ingredients to each group is presented in S1 Table in S1 File.

## Weather data

Local weather patterns have been previously shown to affect butterfly distributions, abundances, and the timing of life history stages, although the strength of such associations varies by species and land cover [25, 43, 47, 68]. To control for potential weather effects on annual butterfly abundance, we generate county-level measures of precipitation and temperature that capture variation between years and within seasons.

Daily weather data were gathered from NASA Daymet, a 1km x 1km spatial grid of daily weather conditions using data from a network of weather stations. We average values over each county's boundaries [44]. Temperature is measured in growing degree days (GDDs), which measures the number of degrees Celsius within a range in which butterflies can develop (11.5˚C to 33˚C). Precipitation is measured in millimeters. We partition each season into three intervals: early (March and April), mid (May and June), and late (July and August). Daily

accumulation of precipitation and GDD are summed over each interval. The resulting variables measure accumulated precipitation and GDDs for each county during each interval for each year.

## Models of butterfly response

We estimate a negative binomial regression model with county and year fixed effects to estimate expected butterfly species richness, total abundance, and *Danus plexippus* (monarch) abundance in each county-year. For county *i* located in CRD $c(i)$ in year *t*, we treat the observed butterfly count ($y_{it}$) as a negative binomial random variable with covariates on the log-link scale,

$$log(y_{it}) = \beta_0 + \beta_P \cdot pest_{it} + \beta_W \cdot weather_{it} + \beta_l \cdot landcover_{it} + \beta_i + \beta_t + log(minutes_{it}). \quad (1)$$

For covariates, we include the vectors of weather variables ($weather_{it}$), land cover ($landover_{it}$), and the vector of pest control group area-treatments ($pest_{it}$). To control for unmeasured temporally invariant factors within each county and spatially invariant factors within each year [55], we also include vectors of fixed effects for county ($\beta_i$) and year ($\beta_t$). Finally, we control for changes in sampling by including the summed duration of all surveys measured in minutes ($minutes_{it}$) as an offset. As a result, the exponentiated dependent variable should be interpreted as the rate of butterflies counted per minute.

We obtain quasi-maximum likelihood coefficient estimates via R 4.0.5 [69]. We compute standard errors robust to violations of the assumed specific relationship between the conditional mean and the conditional variance specified by the negative binomial distribution [55]. We assess model fits via McFadden's pseudo R-squared. We use our robust standard errors [70] to perform z-tests ($\alpha = 0.05$) against the null hypotheses that each coefficient is equal to zero, and to compute confidence intervals for parameter estimates.

## Net pest control effects in counterfactual analysis

To estimate the cumulative impact of all changes in pest control over the study period, we compare predictions in expected values between two pest control use scenarios for each county-year observation in our panel. We compute the difference between predicted population values computed using observed pesticide values ($\hat{y}_{it}$) and predictions computed using pest control values observed for each county in 1998 ($\hat{y}_{it}^{98}$), the first year of the study period. With $f(\cdot)$ as the exponentiated right-hand side of Eq (1), $x_{it}$ as a vector of all covariates other than $pest_{it}$, and $\hat{\beta}$ as the vector of estimated coefficients, these predicted values are calculated as:

$$\hat{y}_{it} = f(pest_{it}, x_{it}; \hat{\beta}); \text{ and} \quad (2)$$

$$\hat{y}_{it}^{98} = f(pest_{i,t=1998}, x_{it}; \hat{\beta}). \quad (3)$$

This difference is divided by $\hat{y}_{it}^{98}$ to compute proportional change:

$$\frac{\hat{y}_{it} - \hat{y}_{it}^{98}}{\hat{y}_{it}^{98}}. \quad (4)$$

This metric is computed for butterfly species richness, total butterfly abundance, and individually for monarchs. We further distinguish between insect control and weed control effects by holding only specific pest control classes at their 1998 values during calculations. We

compute the median net pest control effect across counties for each species in each year to observe trends over time.

## Estimates under alternative assumptions

In support of our GLM estimates, we present a plot of pairwise correlations and Variance Inflation Factors (VIFs) for each explanatory variable (S3 Fig in S1 File). Strong correlation between explanatory variables can induce multicollinearity in GLM estimates that results in inflated variances in coefficient estimates but does not affect the statistical consistency of the estimator [71]. VIFs were fairly low for the pesticide use variables, despite fairly high pairwise correlations between some of these variables. We also present estimates from an alternative version of the panel composed only of counties which contributed butterfly survey data for three or more years (24 fewer counties, N = 607, S4 Table, S4 Fig in S1 File). While the standard errors are larger, coefficient estimates are characteristically similar to those presented below in both sign and magnitude. We present the full dataset because it permits the most rigorous evaluation due to improved statistical power utilizing all available information, with the reduced dataset showing the sensitivity of our results to sample size (S4 Fig in S1 File) and the need for ongoing, comprehensive data for all drivers and responses. We also estimated the mixed model using random effects rather than fixed effects for county and year using the *lme4* package for R [72] (S5 Table in S1 File). The pest control results using this approach were again characteristically similar to those estimated using the preferred GLM approach. Finally, we estimated the base GLMs with only one pesticide class variable included at a time, for six additional estimations for each dependent variable (S6 Table in S1 File). These auxiliary estimates demonstrate the importance of including relevant substitute products when accounting for drivers of biodiversity trends dependent on farmer behavior.

## Results

To disentangle the individual contributions of each hypothesized environmental driver (six pesticide classes, three land-cover types, six weather variables; Fig 2), we interpret effect sizes calculated from the GLM parameter estimates, separately for butterfly species richness and abundance, when all other factors are assumed to be constant at their mean values (Fig 3). Pesticides, particular neonicotinoids and non-glyphosate herbicides, had statistically significant, but mixed, associations with yearly changes in species richness (Fig 3A). They also had weak, non-significant associations with yearly changes in total abundance (Fig 3B). Neonicotinoids had significant, negative associations with monarch abundance (Fig 3C). Weather was the most consistent factor governing year-to-year variability in species richness and abundance; warmer, wetter years were associated with higher species richness (Fig 3C), while cooler, drier years were associated with higher overall abundance (Fig 3F). This was particularly true for monarchs, for which hot temperatures in mid-summer reduced abundance and high precipitation in mid-summer increased abundance. None of the land cover variables were statistically significant at the 95% level (Fig 3B, 3E, 3H).

Building on the GLM estimates, the net effect counterfactual calculations show how predicted abundance and richness vary year-to-year compared to when certain pesticide variables are held constant (Fig 4). By 2014, butterfly abundance in the median county was eight percent lower in projections under observed conditions versus the counterfactual projections where pesticide use was held constant at 1998 levels. Our results show that insecticides, and not herbicides, are associated with negative effects (Fig 4D, 4E, 4F). As with overall abundance, monarchs declined in abundance in the median county by 33% in projections under observed pesticide use levels versus the counterfactual projections (Fig 4I). Although our counterfactual does not distinguish between

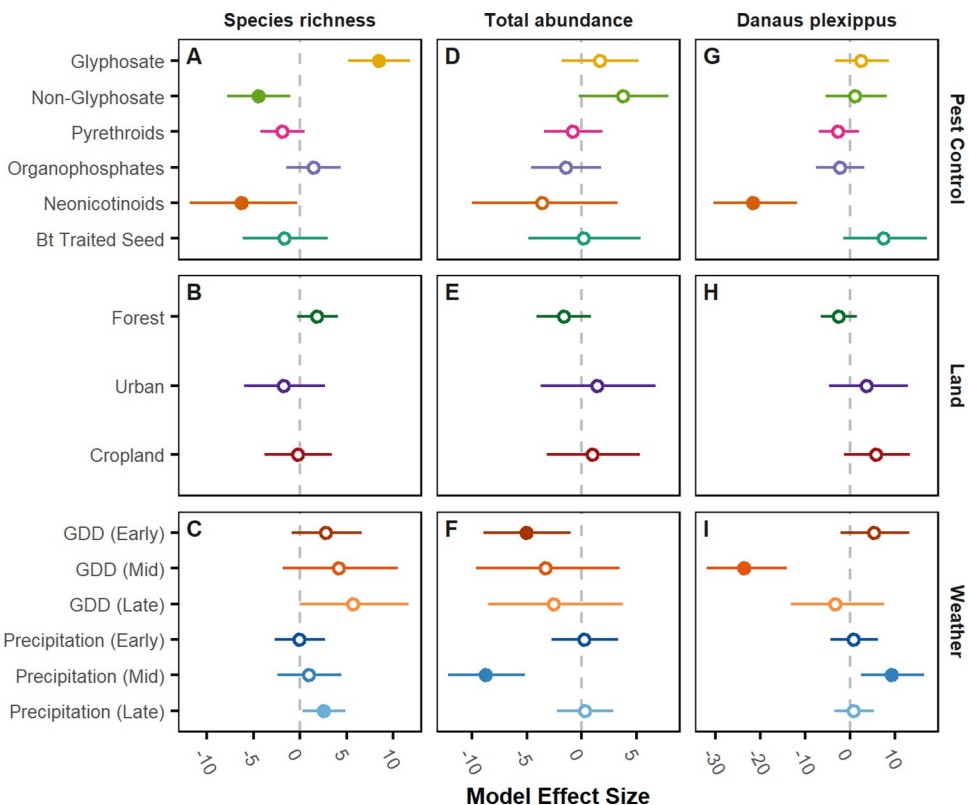

**Fig 3. Model effect sizes for a standard deviation change in explanatory variables.** (A-I) show model effect size estimates from GLM models for the species richness (number of distinct species identified per hour, A-C), total abundance (number of butterflies counted per hour, D-F), and *Danus plexippus* abundance (monarch, G-I). Effect sizes are derived from GLM negative binomial; see Methods for details. For land use, categories are compared to an "open space" baseline. Lines show 95% confidence intervals based on heteroskedasticity-consistent standard errors (HC1). Solid points show statistically significant point estimates at the 95% level. Example interpretation: A one standard deviation increase in neonicotinoid seed treatment area-treatments is associated with 22% lower county-wide abundance for monarchs (G) and 7% fewer distinct species observed (A).

the different classes of insecticides or herbicides (as shown in our GLM effect size results, Fig 3), we note that declines in total abundance and monarch abundance related to insecticide use begin in 2003 (Fig 4E), coincident with the initial deployment and rapid adoption of seed-treated neonicotinoids in corn and soybean plantings in the Midwest (Fig 2H). By contrast, the other two insecticide types saw relatively stable use during the same period (Fig 2I).

As with abundance, insecticides were associated with an 8% decline in butterfly species richness (Fig 4A). Responses to herbicides were more diverse, with species richness observed to be higher than expected and abundance remaining roughly stable compared to the counterfactual over the entire 17-year time series (Fig 4C). Here, we note an observed shift in dynamics for species richness starting around 2008 (Fig 4B, 4E), a breakpoint that coincides with the proliferation of weed resistance to glyphosate [Box 1; 48] that triggered a switch in the trajectories of glyphosate vs. non-glyphosate herbicide use (Fig 2J, [51]). In combination, weed and insect control measures appear to counteract each other, resulting in negligible changes in species richness.

## Discussion

We show that the shift from reactive insecticides to prophylactic tactics has had a strong, negative association with butterfly abundance and species richness in the American Midwest.

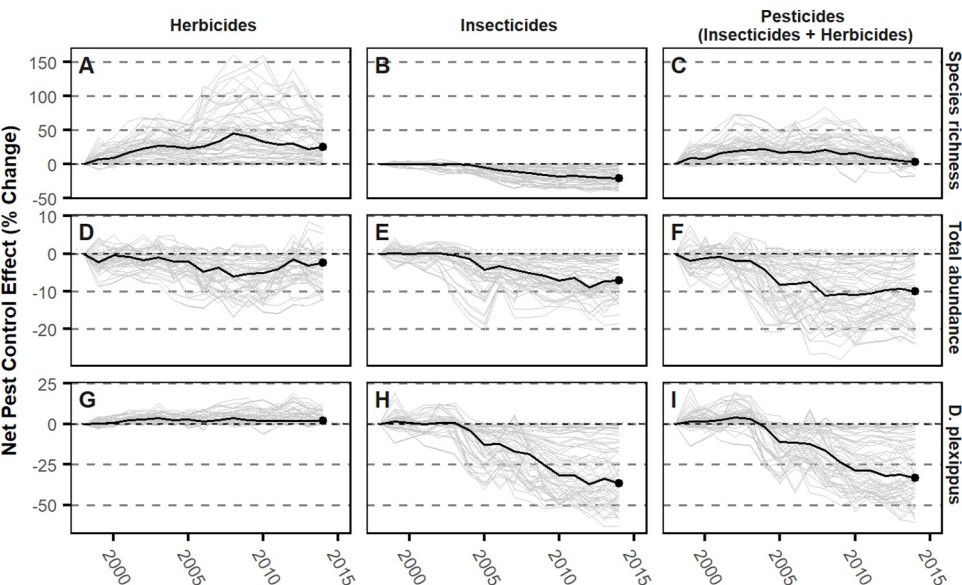

**Fig 4. Net effect of pest control system change over time.** Net pest control effect is the percent change between predicted incidence rates under observed pest control adoption and synthetic counties where pest control adoption is fixed at 1998 levels. County-level net pest control effects over time are shown in gray lines and annual median net pest control effect in black lines; columns show effects computed holding all weed control, insect control, and pest control variables at 1998 levels. Results are presented for species richness (A-C), total abundance (D-F), and *Danus plexippus* (monarch) abundance (G-I).

Taken together, our effect size estimates (Fig 3) and counterfactual simulations (Fig 4) provide different insights into cumulative associations across pesticide classes and their independent relationships, respectively. Our counterfactual analyses show that insecticides account for declines in butterfly species richness and total butterfly abundance over our 17-year study period relative to an alternative future where insecticide use was held constant (Fig 4).

Effect sizes, by themselves, could not explain directional trends over time because they alone do not account for how the entire suite of underlying drivers accumulate through time [73]. Even for individual pesticide classes that are not statistically significant, however, the counterfactual results (Fig 4) show that, in combination, the mean estimated effect of all four insecticide classes are negative, with neonicotinoids having the most negative estimated effect size (Fig 3D). These small, but highly variable effects, that account for farmers' substitution between alternative approaches to pest control, appear in our counterfactual to accumulate to a persistent negative trajectory (Fig 4E). Further, these effects appear to trigger the decline starting in 2003, the year that neonicotinoid-treated soybean seed became available in the Midwestern US (Fig 2E; Box 1). The negative effects of all insecticides on richness appear to be counteracted by changes in herbicide use resulting in a net zero change in species richness (Fig 4C). For total abundance, the effect of insecticides is apparent at least through 2014, the last year for which neonicotinoid data were publicly available (Fig 4B). Notably, the effect sizes estimated with covariates accounting for the full suite of pest control products available to farmers produced considerably different estimates than when only one product was considered at a time, demonstrating the importance of accounting for relevant substitutes as product mixes change (S6 Table in S1 File).

Results in our counterfactual may seem to contrast with the observed trends, which show a decline over time in species richness but not abundance (Fig 2E, 2F). In reality, weather, which has a stronger immediate impact on abundance and richness compared to pesticides (Fig 4, S5

Fig in S1 File), may counteract the impact of pesticide use as it changes over time (Fig 2H–2J). While weather fluctuated over time, it ultimately showed no trend in how it would impact the butterfly community (S5 Fig in S1 File). Through counterfactual analyses that take into account focal factors through time with other drivers held constant, the cumulative effects of multiple changes in drivers can be better deduced.

The negative effects of neonicotinoid penetration to non-crop settings and subsequent uptake in other plants in adjacent habitat [74] contrasts with the relatively weak effects of both i) prophylactic, sprayed insecticides and ii) genetically modified *Bt* seeds. As Midwestern butterflies do not feed on corn, they could only be exposed through *Bt*-contaminated pollen spread to plants in adjacent fields [44]. After a study suggested *Bt*-contaminated pollen spread might be occurring in monarchs, a subsequent in-depth risk analysis showed this not to be the case [44]. Although any quantity substitution of *Bt* seed for sprayed insecticides would be accounted for statistically in our analyses, it is also possible that the use of *Bt* seed actively reduces butterfly exposure by shifting insecticide spraying to earlier in the year [75]. In summary, our results suggest *Bt* seed adoption lacks the harmful effect on butterfly abundance and diversity that is associated with neonicotinoids, supporting the ecological benefits of continued and expanded use, and efforts to protect it from evolved resistance in target pests [76].

In our analyses, the accumulation of sub-lethal effects of pesticides and the potential interactions among many environmental factors that present in variable combinations across space and time make disentangling the impact of any one factor very challenging [1]. This is especially true of pesticides, whose many formulations and methods of delivery are intractable to test in all possible combinations in the lab. For instance, pyrethroids have been found to be among the most potent agents of toxicity for monarchs, much more so than neonicotinoids [77]. Yet, in our analysis, pyrethroids did not show as being as strongly associated with lower monarch abundances or for the larger butterfly community metrics as neonicotinoids (Fig 3A, 3D, 3G). One potential explanation for this is that these chemicals are used reactively and not prophylactically as is done for seed-treated pesticides, so there is less potential for build-up in the environment. Further, interactions with other drivers and/or sub-lethal effects of pesticides have increasingly become the focus in studies of non-target species [78, 79], including bees [80], and also other species of butterflies [81]. Nevertheless, understanding disconnects between laboratory exposure and field surveys should be a focus of future research [50].

The deleterious effects of land use change on biodiversity patterns are widely recognized, but here we show no real impact of land use change. This is because land use did not vary significantly during the timeframe of our study. Indeed, the major land use changes in the majority of the US had already occurred by the early 1900s [21]. On the other hand, we do see a dominant influence of weather on yearly population sizes (Fig 3), which is not surprising because most butterfly species' abundances are known to fluctuate substantially from year to year, and these fluctuations are generally governed by the yearly variability in weather [20]. Succinctly, "good" or "bad" butterfly years compared to other years are likely due to weather conditions that are favorable for caterpillar development and/or the availability and quality of their host plants [82].

These impacts of weather may dampen an observable impact of pesticides during any one year, even while not governing overall trends through time. This is shown clearly by comparing the projections of the counterfactual showing insecticide use accumulating to drive temporal trends of butterflies (Fig 4), whereas weather only causes fluctuations, but no directional trends (S5 Fig in S1 File). However, our study was truncated in 2014, just as a record run of record-breaking heat was tracked across the globe [83].

Our counterfactual analysis indicates that insecticides, and not herbicides, are the strongest pesticide factor associated with monarch declines. When we include all available data,

neonicotinoids are the most significant factor affecting relative local monarch abundances (Fig 3G). It may appear surprising to find no association between glyphosate herbicides and local monarch abundances (Fig 3G), since the impact of glyphosate on milkweed has been well documented [66]. Yet it is important to understand these impacts within the shifting landscape of stressors that have been impacting monarchs for decades. Although glyphosates have been the focus of substantial research and debate around monarch population dynamics for years, a consensus has emerged that glyphosates are likely a dominant cause of monarch declines, but only in the 1990s [50, 84]. Beginning in the early 2000s, that impact largely disappeared since the largest decline in milkweed had already occured [25]. Further, as glyphosate began to become less effective later in the same decade (Fig 2J), that likely further reduced its impacts in the field.

Instead, our study found that neonicotinoids were the most significant environmental driver governing the spatiotemporal distribution of monarchs in the Midwest (Fig 3), even compared to summer weather (S4 Fig in S1 File). Here, we stress that our study only took into account summer weather and the degree to which that drove patterns across the Midwest. Again, this may seem to diverge from recent studies showing climate to be the most important factor driving recent declines in monarchs, but this was climate during spring breeding in Texas [26], a factor we do not take into account. In fact, our study does indicate that hotter temperatures in the middle of summer cause substantial declines, a result that is congruent with past studies [50]. However, during the period of this study there was no directional influence of weather (S5 Fig in S1 File), even though it was an important factor driving spatial differences between counties (Fig 3).

The result that neonicotinoids are negatively associated with monarch abundance is consistent with recent studies of factors affecting the decline of both the western [68] and the eastern [24] populations of monarchs. This result does not align with the relationship between neonicotinoids and monarch mortality in lab toxicology studies; indeed these lab studies show that neonicotinoids are among the least toxic agents [77]. Further, field-relevant exposure levels are estimated to be well below those expected to cause monarch mortality based on laboratory toxicology studies [85, 86]. Yet, the prophylactic use of neonicotinoids means that there is far more opportunity for buildup in the environment, and the true environmental distribution is not known. Further, the sub-lethal impacts of exposure can be difficult to anticipate in real-world settings [87, 88], especially as they interact with other stressors, such as climate. Ultimately, our case study of the monarch reveals how complex these interactions are and how difficult they can be to tease apart in the field. The monarch is certainly one of the best-studied butterflies worldwide [62], and the wealth of information available makes it possible to make sense of these results in the context of its well-known life-history. Unfortunately, this is not the case with most other species, and that makes species-by-species analyses difficult to interpret and why we focused most of our results on community metrics.

Another major concern that arises from this research is that policy formation from studies of effects of modern pesticides on insect decline after 2015 is not possible. Our comprehensive study of multiple drivers across the Midwest US for 17 years ending in 2014 covers a period of rapid technological change in pesticides and crop genetics that has not heretofore been possible. Since 2014, butterflies have continued to decline at the same rate [6, 8]. Insecticidal seed treatment survey data, including the raw data underlying widely available and well used USGS estimates, ceased to be collected after 2014, limiting the ability of the scientific community to expand the scope of these analyses [67]. Further, as revealed by our effect size results for our 17-year study, yearly weather conditions strongly affect butterfly communities (Fig 3C, 3F) but are insufficient to explain the decline in butterfly abundances (S5 Fig in S1 File). However, since neonicotinoid data are not available post-2014 [33], an analysis to compare the

importance of pesticides vs. weather during the seven hottest years since 2014 on record is not possible. Our results expose a need for publicly available, reliable, comprehensive, and consistently reported pesticide use data, particularly for neonicotinoid seed treatments, to fully understand the drivers of butterfly decline.

## Supporting information

**S1 File.**
(DOCX)

## Acknowledgments

We thank Naresh Neupane for extracting climate data; Elise Larsen for aggregating butterfly survey data; Frank Lupi, Doug Landis, Cheryl Schultz; and attendees of the Heartland Environmental and Resource Economics Workshop at the University of Illinois for comments on earlier versions. This research would not be possible without the contributions of hundreds of volunteers and the directors of the programs that coordinate their efforts. We thank Doug Taron of the Illinois BMN (Butterfly Monitoring Network), Jerry Wiedmann of the Ohio BMN, Ronda Spink of the Michigan BMN, and Nathan Brockman of the Iowa BMN, and the hundreds of volunteers who contribute data through these programs. The conclusions and opinions expressed in this article do not necessarily reflect the views or policy of the Washington Department of Fish and Wildlife, the Midwest Climate Adaptation Science Center, or the USGS.

## Author Contributions

**Conceptualization:** Braeden Van Deynze, Scott M. Swinton, Leslie Ries.

**Data curation:** Braeden Van Deynze.

**Formal analysis:** Braeden Van Deynze.

**Funding acquisition:** Scott M. Swinton, David A. Hennessy, Leslie Ries.

**Investigation:** Braeden Van Deynze, Nick M. Haddad, Leslie Ries.

**Methodology:** Braeden Van Deynze.

**Project administration:** Nick M. Haddad, Leslie Ries.

**Supervision:** Scott M. Swinton.

**Validation:** Braeden Van Deynze.

**Writing – original draft:** Braeden Van Deynze, Nick M. Haddad, Leslie Ries.

**Writing – review & editing:** Braeden Van Deynze, Scott M. Swinton, David A. Hennessy.

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
