## [Decision Letter · Decision Letter 0]

12 Oct 2023

PONE-D-23-27327Neonicotinoids, more than herbicides, land use, and climate, drive recent butterfly declines in the American MidwestPLOS ONE

Dear Dr. Haddad,

Thank you for submitting your manuscript to PLOS ONE. After careful consideration, we feel that it has merit but does not fully meet PLOS ONE’s publication criteria as it currently stands. Therefore, we invite you to submit a revised version of the manuscript that addresses the points raised during the review process. The manuscript does not currently meet the PLoS ONE standards for publication #3 (Experiments, statistics, and other analyses are performed to a high technical standard and are described in sufficient detail) or #4 (Conclusions are presented in an appropriate fashion and are supported by the data).   I have provided additional details and discussion of the reports of the reviewers and the manuscript below.

We look forward to receiving your revised manuscript.

Kind regards,

Travis Longcore, Ph.D.

Academic Editor

PLOS ONE

“Naresh Neupane for extracting climate data; Elise Larsen for aggregating butterfly survey data; Frank Lupi, Doug Landis, Cheryl Schultz, and attendees of the Heartland Environmental and Resource Economics Workshop at the University of Illinois for comments on earlier versions. This research was completed while BVD was funded by an NSF LTER grant at Michigan State University. Funding for the Ries Lab was supported by funding from the USGS- Midwest Climate Adaptation Science Center (G21AC10369). This research would not be possible without the contributions of hundreds of volunteers and the directors of the programs that coordinate their efforts.  We thank Doug Taron of the Illinois BMN (Butterfly Monitoring Network), Jerry Wiedmann of the Ohio BMN, Ronda Spink of the Michigan BMN, and Nathan Brockman of the Iowa BMN, and the hundreds of volunteers who contribute data through these programs. The conclusions and opinions expressed in this article do not necessarily reflect the views or policy of the Washington Department of Fish and Wildlife.”

“NSF Long-term Ecological Research Program grant DEB 2224712 (SMS, BVD, NH)

Michigan State University AgBioResearch (SMS)

USDA National Institute of Food and Agriculture (SMS)

Elton R. Smith Endowment for Food and Agricultural Policy (DAH)

5. We note that Figure 1 in your submission contain copyrighted images. All PLOS content is published under the Creative Commons Attribution License (CC BY 4.0), which means that the manuscript, images, and Supporting Information files will be freely available online, and any third party is permitted to access, download, copy, distribute, and use these materials in any way, even commercially, with proper attribution. For more information, see our copyright guidelines: http://journals.plos.org/plosone/s/licenses-and-copyright.

1. You may seek permission from the original copyright holder of Figure(s) [#] to publish the content specifically under the CC BY 4.0 license.

6. We note that Figure 2 in your submission contain [map/satellite] images which may be copyrighted. All PLOS content is published under the Creative Commons Attribution License (CC BY 4.0), which means that the manuscript, images, and Supporting Information files will be freely available online, and any third party is permitted to access, download, copy, distribute, and use these materials in any way, even commercially, with proper attribution. For these reasons, we cannot publish previously copyrighted maps or satellite images created using proprietary data, such as Google software (Google Maps, Street View, and Earth). For more information, see our copyright guidelines: http://journals.plos.org/plosone/s/licenses-and-copyright.

1. You may seek permission from the original copyright holder of Figure 2 to publish the content specifically under the CC BY 4.0 license. 

Additional Editor Comments:

As you may have anticipated, the approach and results presented in this manuscript are polarizing, and led to my expansion of the reviewer number to ensure a fair and thorough review. As you'll see, the reviews range from outright hostility to complete support. I would encourage consideration of criticisms in each of the reviews providing detailed commentary as they get at similar challenges for the reader and the question of whether the conclusions are indeed supported by the data.

There is some question about the methods and exactly how some of the time series contribute to the analysis, especially when there are, for example, only two points in the time series. It would seem that a county with only two surveys would need to be excluded. As it stands, the models are not convincing the reviewers, and will need further explanation (first) and then the question may still remain if they show what they are characterized as showing.

The overall framing and incorporating the breadth of literature in the field on these topics needs to be revisited, especially if the "blame" for monarch decline is going to be shifted now to neonics the literature on actual effects of neonics on monarchs needs to be evaluated as part of describing the purported mechanism that the models are suggesting. That is, drawing conclusions from the model needs to be supplemented by an engagement of the mechanism by which the purported effect is happening and compared with studies that investigate the direct effect of neonics on monarchs.

The manuscript leaves the impression that glyphosate does not impact monarch population trajectories. You discuss that this is probably because the damage has already been done (first paragraph page 8), but this observation needs to take a much bigger role in structuring the narrative overall if this is to be a blame-shifting paper for monarch populations. The argument in the discussion would need to be that "agricultural practices using glyphosate hammered butterfly habitat in agricultural landscapes and got to a point were so much habitat was gone that additional use had no further impact (the earth was already scorched) and now we are seeing that there is a signal that neonics are having a further impact on monarchs on top of the destruction that occurred already." Assuming this is your argument, of course.

As a structural matter, please separate out results and discussion. The ms reads like it started its life at *Science* or *Nature*. For PLoS ONE we aren't dealing with those length and style constraints. Should you prepare a revision, it would be appropriate expand the introduction and include a background literature review to address the holes identified by reviewer 1, go into more detail on the methods, report the results separately and without commentary, and then provide a discussion and conclusions that meets PLoS ONE standards ("[A]uthors should avoid overstating their conclusions. Authors may discuss possible implications for their results as long as these are clearly identified as hypotheses instead of conclusions.").

I am requesting a major revision of the manuscript, but please note that does not guarantee that it will be accepted following revision.

An engagement of the comments of each of the reviewers is requested, especially reviewers 2 and 4 relevant to the methods. Reviewer 1 was convinced of the fatal weakness of the manuscript and although pointed out areas where additional context and background information is necessary, did not deeply engage the methods. The "big picture" critiques from reviewer 1 do need to be substantively addressed, however.

Reviewers' comments:

Reviewer's Responses to Questions

**Comments to the Author**

1. Is the manuscript technically sound, and do the data support the conclusions?

Reviewer #1: No

Reviewer #2: Yes

Reviewer #3: Yes

Reviewer #4: Partly

2. Has the statistical analysis been performed appropriately and rigorously? 

Reviewer #1: No

Reviewer #2: Yes

Reviewer #3: Yes

Reviewer #4: I Don't Know

3. Have the authors made all data underlying the findings in their manuscript fully available?

Reviewer #1: Yes

Reviewer #2: No

Reviewer #3: Yes

Reviewer #4: Yes

4. Is the manuscript presented in an intelligible fashion and written in standard English?

Reviewer #1: Yes

Reviewer #2: Yes

Reviewer #3: Yes

Reviewer #4: Yes

5. Review Comments to the Author

Reviewer #1: General

This manuscript describes an analysis of multiple butterfly monitoring programs in the American Midwest, over a number of years, and attempted to ascertain if any anthropogenic factors such as pesticides, herbicides, landuse, or climatic variables, influence either the diversity of butterfly species or their abundance. They evaluated all monitored species, but highlighted the effects on monarchs, because (I think) of their popularity, and because of the ongoing narrative about their population trends. There were a number of significant trends with some the variables tested, though the authors highlighted only one of these, which was a negative effect of neonicitinoid pesticides on the species richness, and also on the abundance of monarchs.

I have to say that I was initially very intrigued by the idea of this study, but after reading this paper, I am rather shocked at so many instances where the authors appeared to go out of their way to either mischaracterize or purposely ignore the latest science around butterfly and insect populations here in North America, plus some recent research on monarch butterflies (including their response to neonicitinoids), and even some prior work by one of these labs. I find this very troubling, and it does not speak well of the leadership of the senior authors here. Further, the results of these analyses here seem to be counter to much of the established science just mentioned, which calls the data or analyses into question. If this paper is to go forward (either in this journal or after rejection, to another) then the authors need to do a gut-check, and think hard about how this study will affect their reputations.

Specific points

There is so much to point out about this paper that it is hard to know where to start. But for now, let me point out the parts/statements that are mischaracterized. First, the introduction begins with a loaded statement that all insects are declining, which is not entirely true, at least in North America. Indeed, much of this paper is built around this story of insects (butterflies) being in trouble. There have certainly been analyses of long-term data that shows declines of some insects, but then increases in others (Van Klink et al 2020). There has been work showing declines of butterflies in the west (Forister et al 2021) but then a more recent study (Crossley et al 2021) showed this decline was specific to the west, and the butterflies in the east were improving. Another of Forister’s papers showed that some butterfly species are “winning” in the Anthropocene. I note that these studies showing good news were distinctly missing from the text of this paper. Really, the situation is not a straightforward “decline of all insects”, at least in North America.

Along the same vein, the paper has much language devoted to describing the monarch butterfly as being in trouble, which is also a mischaracterization, based on the latest research (which was also left out here). Some of this research even was authored by one of the authors here. Ries et al 2015 described an analysis of long-term monitoring data from another program, which showed no decline of monarch abundance. Similarly, Inamine et al 2016 described a similar analysis of even more data. Crossley et al. 2022 had probably the most comprehensive analysis of monarch data to date, which again, showed no long term decline in the number of summertime monarchs across the continent. Then, the latest study of the monarch genome (Boyle et al 2023) describes how there has been no long-term decline in the effective population size in the last 75 years. Further, it now looks like the population size now is essentially larger than it ever was prior to human clearing of the prairies and eastern forests 200 years ago. So, the story that monarchs are in decline (even though it is popular in the media) is not accurate based on the latest science. I also note that none of the studies I mentioned were brought up in this paper. I find it hard to believe that this was a simple case of carelessness. More likely the authors did not want to include them, which is highly inappropriate.

The focus here on the Midwest only did not help to boost my view of this paper either, as it can lead to misleading conclusions. As the Crossley et al 2022 study demonstrates, the butterflies in the U.S. are experiencing declines in some regions, increases in others, and are stable in lots more. So, taking a snapshot of one region (even with a boatload of data) is still just one region, and so the conclusions from this analysis would really only be pertinent to here (i.e. probably could not be extrapolated). Moreover, even focusing on the monarchs alone in this region is problematic. The authors state that this is appropriate for monarchs since "95% of monarchs that reach the winter colonies originated in the cornbelt" (citing Wassenaar and Hobson 1998). This is an outdated conclusion, and again, a mischaracterization because the proper (more recent) science was not even mentioned. Wassenaar and Hobson only examined one collection of monarchs from 1997 (an exceptional year). A more thorough study was done by Flockhart et al 2017, in which 40 years of samples were examined. Their results highlighted that the Midwest only produces about 37% of the monarchs that reach Mexico, while 63% come from elsewhere. In fact, their results showed that this varies by year – in some years, most monarchs that reach Mexico come from the northeast, and in others it is the far north. Again, a seeming lack of understanding of the science is the problem here, or perhaps a purposeful effort to neglect the important studies.

I’m going to jump right into the main conclusions next for the sake of my time, since there is just too much to go through line-by-line. The main conclusion of this analysis is that neonicitinoid pesticides are causing declines to butterflies, and to monarchs in particular. This is a result that sounds ridiculous, based on the accumulated laboratory research thus far. The effects of neonics on monarchs specifically have been assessed multiple times, by multiple labs, and each time, there is little to no effect found, at least when standard, field-relevant doses are used. For some reason, monarch butterflies are resistant to nearly every neonicitinoid chemical tested, likely because of their ability to take up the toxins from milkweeds. In a few cases, declines have been found, but only when unrealistic doses are given. Further, multiple labs have examined if the plant concentrations of these chemicals are high enough in the field to cause harm to monarchs, and the answer there is also no. Off the top of my head, I can think of the following studies: Prouty et al 2023, Hall et al 2022, Bargar et al 2020, Olaya-Arenas and Kaplan 2019. None of these studies were mentioned at all. Again, an egregious oversight.

There was another result of these analyses that seemed to be purposely overlooked by the authors, but yet would have tremendous implications if this paper were ever to see the light of day. Apparently, there is a statistically significant, positive effect of glyphosate use on butterfly diversity, and at least a positive trend on monarch abundance (Fig. 4A). This result seems to strain credulity. Not only that, the authors here found NO negative effect of glyphosate use on monarch abundance, even though multiple prior studies (even by these authors) have apparently concluded that glyphosate use in the Midwest was destroying monarch habitat, and leading to their demise.

Yet another important (and credulity-straining) result is that Bt-treated plants are improving the abundance of monarch butterflies, according to these analyses. Yet again, here is a result that blows a giant hole into a suite of prior (older) research that suggested that this crop modification would harm monarchs. This research was from the early 2000s, and some of that hoopla later died down because it turns out the monarchs were never exposed to as much contaminated anthers as people thought. But, this new analysis seems to suggest that Bt-corn is even good for monarch abundance?

So with this new paper, the authors want the reader to believe that glyphosate is NOT harming monarch numbers in the Midwest, and if anything, is helping to improve butterfly diversity there! So is Bt-corn! AND, the authors want people to believe that neonicitinoid pesticides are now to blame for all of the woes of butterflies, and especially monarchs (which all lab studies have shown are resistant to neonics). This is why I think the authors need to do some soul-searching here, and decide if they really want to damage their reputation(s) with this paper. By publishing this, the authors will essentially be undermining much of all of this the prior work, which will call into question either this paper, or, the validity of prior analyses. And I haven’t even begun to discuss what would happen if the media caught wind of these new results. The media hoopla over these “new” findings would cause an upheaval and undermine public faith in scientists.

I’m going to assume that this journal will reject the paper, and so these comments are really more for the authors to consider, for gauging whether to resubmit elsewhere. This is probably the harshest review I’ve ever written, but it is because the level of arrogance here is palpable – both for purposely misleading the reader on key issues, and for presenting results that are so at odds with so much recent and established evidence, without even as much as mentioning that evidence. The lack of objectivity here is shocking. Please consider what I said. If I were you, I would table this project. It will be a hot potato if published elsewhere.

Reviewer #2: The paper is very well written. The analysis goals of evaluating different drivers of butterfly abundance are interesting and valuable. The dataset is opportunistic but nonetheless well put together (more on that). It covers a useful number of years and locations (more on that). The analysis is well done but could use a few more components for readers to understand possible confounding (more on that). The conclusions are reasonable and are not overblown given the data and analysis methods (more on that). The tables and figures are great. I only have a few thoughts on how to make the paper more convincing (or not).

First, I like the counterfactual analysis as a way of exploring the implications of the model. Given it’s complete dependence on the model, a reader must be convinced that you got the model right if you want them to trust your conclusions. This is a big task in this context, where you have an observational study, two somewhat-vague community-level response variables, a cobbled together dataset, study counties popping in and out of the dataset, and likely loads of correlation among the independent variables. So, here are some thoughts.

First, related to the community metrics. It would be nice to know which species are driving them. Consider creating a plot or table that gives the median abundance per species, like a rank-abundance bar plot. Readers might like to know if the abundance metric is entirely driven by cabbage whites, for example. Or maybe diversity depends entirely on prairie-dependent species. All this will help a reader understand your results better.

Second, regarding an opportunistic dataset, is there geographic or temporal confounding going on? Can you make something like a map that shows when the different counties enter the dataset? Maybe it is like fig 2D? Or maybe it is a table? With 60 rows and columns summarizing the extent of different data types included? And, by the way, does having a county with 1 year of data really help for this analysis? Or is it more likely to cause unexpected weirdness? Related to this, seems like random effects for counties and years would be better than fixed effects. Maybe using fixed effects is an econometrics thing? Related to spatiotemporal confounding, you probably know that it’s good practice on these types of large-scale and -extent projects to look for spatial and temporal autocorrelation in model residuals. Think Luke Anselin.

Third, I imagine there is plenty of correlation among your independent variables. Maybe it is not enough to cause computational problems (extreme multicollinearity). But it is probably enough to cause interpretation challenges. In this case, it is helpful to run univariate correlations or regression models and note the model coefficients. It sure is convenient, and ultimately more convincing, when the univariate effect estimates are similar to the multivariate effect estimates. When that is not the case, due to independent variable correlations, you have to be extra careful in interpreting model results, right?

Why all this annoying model scrutiny? Because if your model is not a slam dunk then any counterfactual analysis, and subsequent conclusions, and subsequent someday ag or environmental policies, and subsequent economic impacts on real people, are not well founded. You could probably publish this paper without any of this additional junk. But if you do the extra work and still come to the same conclusions, then your conclusions will be that much more powerful and valuable. From a less lofty perspective, the monarch mafia is going to have a hard time believing that monarch decline is more about seed coat treatment than glyphosate application or loss of hedgerows due to corn ethanol. So, you’ll need to convince them your results are not confounded by correlations between independent variables.

Reviewer #3: I enjoyed reading this manuscript; it is rigorous and important piece of work. In particular, the scientific approach of contrasting alternative predictors for a problematic species’ (in this case, declining) population abundance is critical to get past simple correlations. Much past work on the focal species examined here has not taken this approach and provided on parts of the story, some of which suffered from correlations. Finally, the authors present a complete and balanced picture. I particularly appreciated the distinction of insecticides vs. herbicides in this study. Finally, the 17 year scope of the work, the use of multiple abundance databases, and the diversity of butterflies covered makes the study of exceptional importance.

Although I am not familiar with the specifics of counterfactual analysis, I did find the data summary, statistics, and reporting convincing.

There are clear policy implications of this work.

Figures are excellent.

Reviewer #4: Review of “Neonicotinoids, more than herbicides, land use, and climate, drive recent butterfly declines in the American Midwest”

This is an ambitious study of existing data, attempting to demonstrate a spatio-temporal correlation of butterfly richness, numbers, and specifically monarch numbers with various factors, including different aspects of herbicide and pesticide use. The authors draw a strong conclusion about the negative effects of neonicotinoids based on the analysis. Given the inevitable controversy that monarch research, especially insecticide-related, generates, there must be more clarity in the methods and discussion of uncertainties for this to be an effective contribution.

Extracting trend data from butterfly monitoring programs can be tricky, although strong signals usually come through the noise. Time series have many factors correlated with time itself, and the combination of counterfactual analysis and the negative binomial GLM attempts to disentangle the factors.

But, the following aspects of the analysis need clarification:

A key figure would be the raw time series of species richness, overall abundance, and monarch counts with the same format as in Figure 2 – a mean/median line with the cloud of county level data.

The period 1998-2014 had data from 78 counties (SI table). The abstract says 60 counties. Which 60? (in the S.I Table). Is it the counties with at least 2 years? Is that how many faded lines there are in each graph in Figures 2 and 3? It is not clear how the interrupted time series of county-level counts were handled – 28 counties had only 1-4 years of data, while 13 had complete time series. The inclusion of the “Year” and “County” co-variables in the GLM (maybe?) addressed this - i.e., factors well beyond the study region, like the 2011 drought in Texas and winter storms in Mexico in the “Year” factor, and the samplingincompleteness in the "County" factor, but much more explanation needs to be made, especially the counties that had 1-4 (or 2-4) years data. Could be in the SI.

In trying to interpret the results, is it correct say that the leverage of each factor is dependent on the spatial variability among counties in, say, insecticide treatments, and the sharp increase in neonicotinoids from 2003-2009?

A very recent paper (not available when this MS was written) has some methods for splitting the time series into periods corresponding to changing factors like herbicide use, since the elimination of within field milkweeds was nearly complete by 2006. The discussion on p.14 of glyphosate hints at this break, but more explicit consideration

(Pleasants, J., Thogmartin, W.E., Oberhauser, K.S., Taylor, O.R. and Stenoien, C., 2023. A comparison of summer, fall and winter estimates of monarch population size before and after milkweed eradication from crop fields in North America. Insect Conservation and Diversity.)

The inclusion of the annual weather data by county is an important control. But the lack of post 2014 pesticide data prohibits consideration of the more recent hottest years (as mentioned on p. 14). Given that neonicotinoids leveled off after 2009, could a similar counterfactual analysis of climate impacts since 2014 reveal more about the weather impacts?

And some of the other significant results from the GLM deserve more discussion (like species richness-glyphosate positive)

The proportion of butterflies in Mexico from this region should be reported as a range, rather than a single value (P.13). It does change from year to year, but there is no doubt that this region serves as a major source of migratory monarchs.

The published literature on experimentally determined lethal and sublethal effects of neonicotinoids is confusing; some studies show minimal or no effects at field relevant concentrations, other show important mortality and sublethal effects. While neonicotinoids (and other insecticides) undoubtedly kill their targets, it is important to acknowledge the disparate results.

6. PLOS authors have the option to publish the peer review history of their article (what does this mean?). If published, this will include your full peer review and any attached files.

Reviewere#1: No

Reviewer #2: No

Reviewer #3: No

Reviewer #4: No

---

## [Author Response · Author response to Decision Letter 0]

26 Feb 2024

Response to reviews

(our responses are in blue font)

Additional Editor Comments:

As you may have anticipated, the approach and results presented in this manuscript are polarizing, and led to my expansion of the reviewer number to ensure a fair and thorough review. As you'll see, the reviews range from outright hostility to complete support. I would encourage consideration of criticisms in each of the reviews providing detailed commentary as they get at similar challenges for the reader and the question of whether the conclusions are indeed supported by the data,

There is some question about the methods and exactly how some of the time series contribute to the analysis, especially when there are, for example, only two points in the time series. It would seem that a county with only two surveys would need to be excluded. As it stands, the models are not convincing the reviewers, and will need further explanation (first) and then the question may still remain if they show what they are characterized as showing.

As we discuss below in our responses to reviewers, we have conducted extensive new analyses to address this and other comments about analyses. As to this comment, our original analysis is appropriate, and we maintain that, given the way our model was designed, it is appropriate and strengthens the analysis by leveraging all available data. However, we have conducted an additional analysis that removes data from counties that have three or fewer butterfly surveys.

The reason the full dataset is appropriate is that the GLMs are not an analysis of individual time series. Rather, the independent variables are levels in a given county, and the response variables are butterfly abundance or diversity, in each given year. Thus, it is the full complement of counties in each year that permit us to determine the effects of a level of an independent variable in that year. Take an extreme example: if an extreme drought caused there to be zero butterflies in one year in some counties, and if butterfly abundances in one county were measured only once (when there was extreme drought), that would still give information to inform the model in context of all the other counties for which we have data. We explain the rationale for the full and partial datasets in the Methods (lines 368-371). Inevitably, the reduced dataset had lower statistical power so the results are often not significant but, importantly, the patterns are generally similar (Figure S4). 

The overall framing and incorporating the breadth of literature in the field on these topics needs to be revisited, especially if the "blame" for monarch decline is going to be shifted now to neonics the literature on actual effects of neonics on monarchs needs to be evaluated as part of describing the purported mechanism that the models are suggesting. That is, drawing conclusions from the model needs to be supplemented by an engagement of the mechanism by which the purported effect is happening and compared with studies that investigate the direct effect of neonics on monarchs.

The manuscript leaves the impression that glyphosate does not impact monarch population trajectories. You discuss that this is probably because the damage has already been done (first paragraph page 8), but this observation needs to take a much bigger role in structuring the narrative overall if this is to be a blame-shifting paper for monarch populations. The argument in the discussion would need to be that "agricultural practices using glyphosate hammered butterfly habitat in agricultural landscapes and got to a point were so much habitat was gone that additional use had no further impact (the earth was already scorched) and now we are seeing that there is a signal that neonics are having a further impact on monarchs on top of the destruction that occurred already." Assuming this is your argument, of course./

We appreciate that we may not have sufficiently emphasized this apparent disconnect that may arise because of the seeming contrast of our results with past studies. We have added substantial language to explain the historical understanding of causes of monarch decline to put our results in context, both in Box 1 and throughout the discussion (lines 529-572). Importantly, we did not mean for this to come off as “blame shifting” because, as you note, we highlight that we are missing the time frame of the dominant loss of milkweed in the landscape. Indeed, our analyses add the nuance that attributes decline to multiple mechanisms, not just neonicotinoids (as seems to be suggested by our results). We also provide historical perspective on pesticide use that explains why, over time, different drivers may be more important (see Box 1). We address these important issues about the monarch in the revised text and in our responses below.

As a structural matter, please separate out results and discussion. The ms reads like it started its life at Science or Nature. For PLoS ONE we aren't dealing with those length and style constraints. Should you prepare a revision, it would be appropriate expand the introduction and include a background literature review to address the holes identified by reviewer 1, go into more detail on the methods, report the results separately and without commentary, and then provide a discussion and conclusions that meets PLoS ONE standards ("[A]uthors should avoid overstating their conclusions. Authors may discuss possible implications for their results as long as these are clearly identified as hypotheses instead of conclusions.").

As noted above, we have restructured the revision, including moving content from the supplements to the main text, to more closely adhere to the PLoSONE format. 

I am requesting a major revision of the manuscript, but please note that does not guarantee that it will be accepted following revision. An engagement of the comments of each of the reviewers is requested, especially reviewers 2 and 4 relevant to the methods. Reviewer 1 was convinced of the fatal weakness of the manuscript and although pointed out areas where additional context and background information is necessary, did not deeply engage the methods. The "big picture" critiques from reviewer 1 do need to be substantively addressed, however.

We address every comment of reviewer 1 in our response. This review is, in our experience, unique: we have never read a reviewer that is filled with such personal attacks while being almost entirely devoid of any critique of the data or analysis. We address every scientific criticism with a response, including with details of where we addressed a comment with a revision in the manuscript. We address other criticisms as well; coauthor Ries does not want to leave the personal and deeply insulting accusations unanswered. 

Comments to the Author

Reviewer #1: This manuscript describes an analysis of multiple butterfly monitoring programs in the American Midwest, over a number of years, and attempted to ascertain if any anthropogenic factors such as pesticides, herbicides, landuse, or climatic variables, influence either the diversity of butterfly species or their abundance. They evaluated all monitored species, but highlighted the effects on monarchs, because (I think) of their popularity, and because of the ongoing narrative about their population trends. There were a number of significant trends with some the variables tested, though the authors highlighted only one of these, which was a negative effect of neonicitinoid pesticides on the species richness, and also on the abundance of monarchs.

I have to say that I was initially very intrigued by the idea of this study, but after reading this paper, I am rather shocked at so many instances where the authors appeared to go out of their way to either mischaracterize or purposely ignore the latest science around butterfly and insect populations here in North America, plus some recent research on monarch butterflies (including their response to neonicitinoids), and even some prior work by one of these labs. I find this very troubling, and it does not speak well of the leadership of the senior authors here. Further, the results of these analyses here seem to be counter to much of the established science just mentioned, which calls the data or analyses into question. If this paper is to go forward (either in this journal or after rejection, to another) then the authors need to do a gut-check, and think hard about how this study will affect their reputations.

Authors response: As noted above, we address every specific comment of reviewer 1 below, but here we particularly object to the offensive final statement of this paragraph, which reads as a personal attack and, crucially, is not at all supported by any criticism of our data or analyses. 

Specific points

There is so much to point out about this paper that it is hard to know where to start. But for now, let me point out the parts/statements that are mischaracterized. First, the introduction begins with a loaded statement that all insects are declining, which is not entirely true, at least in North America. Indeed, much of this paper is built around this story of insects (butterflies) being in trouble. There have certainly been analyses of long-term data that shows declines of some insects, but then increases in others (Van Klink et al 2020). There has been work showing declines of butterflies in the west (Forister et al 2021) but then a more recent study (Crossley et al 2021) showed this decline was specific to the west, and the butterflies in the east were improving. Another of Forister’s papers showed that some butterfly species are “winning” in the Anthropocene. I note that these studies showing good news were distinctly missing from the text of this paper. Really, the situation is not a straightforward “decline of all insects”, at least in North America.

Author response: We see that in the limited space of the original abstract, our statement that “insects are in decline globally” could be read as “all” species of insects, but we were instead referring to total abundance. In our revised abstract, we have clarified that we are referring to total abundance. However, in the very first sentences of our original introduction, we wrote that: “Globally, declines of abundance and biomass for many insect groups have been reported at rates of 2-4% per year, rates that compound to as much as 30-50% of total abundance loss, depending on study duration.” Further, in lines 20-21 on page 2 of our original submission, we wrote: “... taxon-wide studies for insects show species respond variably, depending on their thermal niche requirements, producing climate “winners” and “losers”.” Therefore, It is difficult for us to grapple with this accusation. We maintain that our original language had appropriate balance; however, we see that if readers go into the main text of the paper with that misconception, then that will color the way they read the entire paper. Thus, in addition to clarifying our statement in the first sentence of the abstract, and retaining the statements above, we also emphasize this point and add more details in the introduction. We report on papers (lines 39-41) that do not show declines, but also point out that these studies are outliers in the larger literature. For example, while van Klink, et al. (2020) does show an increase in the abundance of aquatic insects, there is an overall decrease in numbers of terrestrial insects, which is much more relevant to our study of butterflies. 

Along the same vein, the paper has much language devoted to describing the monarch butterfly as being in trouble, which is also a mischaracterization, based on the latest research (which was also left out here). Some of this research even was authored by one of the authors here. Ries et al 2015 described an analysis of long-term monitoring data from another program, which showed no decline of monarch abundance. Similarly, Inamine et al 2016 described a similar analysis of even more data. 

Our abstract echoes reviewer 1’s comment, as it reads: “This included the abundance of the migratory monarch (Danaus plexippus), whose decline is the focus of intensive debate and public concern.” As we are consistent with reviewer 1’s sentiment, but contrary to reviewer 1’s comment that we describe “monarchs being in trouble,” we cannot address this comment in any depth in our manuscript. The monarch literature is vast, so much so that it cannot be summarized in this paper. One of us just published a comprehensive summary of population trajectory and driver studies (Shirey and Ries 2023), and that paper’s 8000 words address just one branch of the monarch literature. Importantly, this review shows that the bulk of evidence in the most rigorous studies shows strong population declines in the dominant breeding grounds of both east and west populations. In our resubmission, we cannot address most of reviewer 1’s points because they are outside the scope of this study. However, we did expand on our language, adding 3 paragraphs to emphasize more about the history of monarch studies, so we can more clearly provide readers with more context about monarch biology (lines 529-572). 

To add detail to reviewer 1’s reference to “the author’s own” paper: Ries et al.(2015) and, similarly, Inamine et al., (2016) showed no decline in monarch abundance indices on the summer grounds using NABA and IL data. Thus, the papers that reviewer 1 cites suggested a “disconnect” between the summer (no trend) and winter (declining trends) data (Ries et al., 2015; Inamine et al., 2016). However, in both studies, declines in Mexico were clear and subsequent studies have also shown declines during summer (Zylstra et al., 2021). Indeed, all studies of the overwinter colonies published since 2012 have shown declines both in the eastern and western populations (Shirey and Ries, 2023). Because the winter aggregations provide a snapshot when the entire population is aggregated in one small region, monitoring of the winter colonies is considered the best stage at which to examine the trajectory of the population size over time (regardless of variability during different stages of the migratory cycle). 

Crossley et al. 2022 had probably the most comprehensive analysis of monarch data to date, which again, showed no long term decline in the number of summertime monarchs across the continent. 

Here, we address what could be construed as a disconnect between the results of a recent publication of monarch population trajectories in summer and winter ranges (especially Zylstra et al., 2021) with the results of Crossley et al. (2022). That study used unweighted averages of population trajectories across the US no matter how big or small the regional monarch populations are, and we find that to be a fatal flaw in their conclusion of no overall decline. This issue was treated in Shirey and Ries 2023 and we quote our summary from Shirley and Ries (2023; see also figure 2 in Crossley, et al. (2022) in support of our interpretation): 

“This is the most comprehensive study showing that across the entire North American range of the monarch, they do not appear to be declining. However, we note that if the spatial patterns of declines, increases, and areas with no apparent trends are examined in their results, there does appear to be a trend towards declines in the Midwest and Northeast, which encompass the main recruitment grounds of the overwinter population in Mexico. Thus, we feel that, despite the interpretation of the authors, our opinion is that this paper actually presents evidence that corroborates our conclusion of monarch declines, at least for the Eastern population.”

Then, the latest study of the monarch genome (Boyle et al 2023) describes how there has been no long-term decline in the effective population size in the last 75 years. Further, it now looks like the population size now is essentially larger than it ever was prior to human clearing of the prairies and eastern forests 200 years ago. 

This comment is beyond the scope of our paper that addresses factors causing changes in population size in recent decades. We point to Shirey and Ries (2023) that acknowledges the Boyle, et al. (2022) study that purports to reconstruct genetic sig

---

## [Editor Report · Decision Letter 1]

10 May 2024

Insecticides, more than herbicides, land use, and climate, are associated with declines in butterfly species richness and abundance in the American Midwest

PONE-D-23-27327R1

Dear Dr. Haddad,

We’re pleased to inform you that your manuscript has been judged scientifically suitable for publication and will be formally accepted for publication once it meets all outstanding technical requirements.

Kind regards,

Travis Longcore, Ph.D.

Academic Editor

PLOS ONE

Additional Editor Comments (optional):

This revision addresses comments from reviewers who ranged from "publish as is" to "burn the manuscript" in detail and does so thoroughly. I have read and in some cases re-read all of the critiques and responses and believe that the manuscript meets the PLoS ONE criteria for publication. Namely that this is 1) original research, 2) reports results not previously published, 3) contains analyses performed to a high technical standard and described in sufficient detail, 4) contains conclusions supported by the data without overstatement (here I appreciate the careful discussion of the limitations inherent in the time period of the data and the effort to preempt misinterpretation by readers), 5) written well in English, and 6) meeting standards of research ethics and integrity. During proofs, please be sure to update the information about data availability with the appropriate repository.
---

## [Editor Report · Acceptance letter]

30 May 2024

PONE-D-23-27327R1 

PLOS ONE

Dear Dr. Haddad, 

I'm pleased to inform you that your manuscript has been deemed suitable for publication in PLOS ONE. Congratulations! Your manuscript is now being handed over to our production team.

Kind regards, 

on behalf of

Dr. Travis Longcore 

Academic Editor

PLOS ONE